# Optimization of Extrusion Treatments, Quality Assessments, and Kinetics Degradation of Enzyme Activities during Storage of Rice Bran

**DOI:** 10.3390/foods12061236

**Published:** 2023-03-14

**Authors:** Muhammad Tayyab Rashid, Kunlun Liu, Simeng Han, Mushtaq Ahmed Jatoi, Frederick Sarpong

**Affiliations:** 1College of Food Science and Engineering, Henan University of Technology, Zhengzhou 450001, China; trashid208@gmail.com (M.T.R.);; 2School of Food and Strategic Reserves, Henan University of Technology, Zhengzhou 450001, China; 3Department of Botany, Shah Abdul Latif University, Khairpur 66020, Sindh, Pakistan; 4Value Addition Division, Oil Palm Research Institute, Council for Scientific and Industrial Research, Kade P.O. Box 74, Ghana

**Keywords:** extrusion, electrophoresis, lipids, protein, stabilization, rice bran

## Abstract

Over the years, extrusion has been a multi-step thermal technique that has proven to be the most effective process to stabilize rice bran (RB). This study aimed to investigate the effects of extrusion treatment and temperature (15, 25, and 40 °C) on the storage stability, lipid oxidation, peroxidase, and peroxide values, free fatty acids, fatty acid composition, and protein variations of RB over 60 days. The study offers novel insights into the changes in RB’s protein and amino acid compositions during extrusion and storage, which has not been extensively explored in prior research. After extrusion processing, peroxidase activity (POD) and lipase activity (LPS) were significantly reduced. However, peroxide value (PV), free fatty acids (FFA), and malondialdehyde content (MDA) observed a significantly increased by 0.64 mEqO2/kg, 8.3 mg/100 g, and 0.0005 μmol/L respectively. The storage stability of RB after extrusion shows that the POD, LPS, FFA, PV, and MDA were positively correlated with storage duration and temperature. The oleic acid/linoleic acid ratio in processed RB by extrusion had no significant changes during storage. The total and essential/non-essential amino acid ratios showed a downward trend of 5.26% and 8.76%, respectively. The first-order kinetics was the best-fitting model to describe the enzymatic inactivation and degradation of extruded RB during storage. The extrusion treatment did not affect the crude protein and the essential subunits of protein. Overall, the optimized extrusion procedure exhibited promising results in stabilizing the RB.

## 1. Introduction

Rice bran is an underrated but nutritionally essential by-product from rice milling and a potential dietary ingredient to promote health [1]. It contains 23–28% crude fibre, 15–20% fat, 7–8% ash, 12–16% protein, minerals, vitamins, phenolic compounds, and essential unsaturated fatty acids [2]. Rice bran contains vital nutrients such as γ-oryzanol, tocopherol, and tocotrienol, which have been shown to protect against cancer, obesity, dyslipidemia, inflammation, and other disorders. It has been demonstrated to decrease cholesterol in both people and experimental animals and reduce the risk of developing cardiovascular disease [3].

Despite its health advantages, RB is mainly utilized only in animal feed. However, its utilization for human consumption is limited due to the problem of rapid rancidity during storage. Developing effective methods for stabilizing rice bran through extrusion treatment could not only reduce waste and increase efficiency in the rice industry but also provide a new source of revenue through the sale of stabilized rice bran for use in the food industry, potentially generating millions of dollars in economic value [4]. As RB includes a lot of lipids, it is highly vulnerable to rancidity during storage. There are hydrolysis and oxidase enzymes, primarily lipase, mainly responsible for triglycerides breaking down to glycerol and free fatty acids (FFA). It must be inhibited shortly before use to avoid fatty acid absorption, increase shelf life, and for human utilization. As a result, it is required to stabilize RB using proper procedures to increase its shelf life under environmental circumstances without compromising its nutritional quality.

There are numerous methods for RB stabilization have been established, including infrared [5], microwave [6,7], radio frequency [8], heating treatment [9,10], and protease [11]. Extrusion heating or cooking is one of the most effective and popular methods of stabilizing rice bran [12]. Extrusion is the process of thermoplastic moulding mashed starch and protein ingredients in an extruder. The raw ingredients are pushed, combined, ground, and squeezed by revolving screws in the extruder’s hot-pressing chamber. When friction and pressure are increased, the temperature of the material quickly rises, reaching 185 °C, causing it to plasticize [13]. Extrusion treatment can efficiently inactivate enzyme function because of high pressure, shear force, and temperature, which can potentially disrupt the protein’s high-level structure [14]. Studies on the application of extrusion treatment regarding the extraction rate and quality of rice bran oil are frequently available. However, the information on protein modifications and lipid oxidation of extruded RB during storage is yet to be fully explored and hence taken in hand in the current study.

Product qualities of extrudates vary significantly based on extrusion processing factors such as mould shape, particle size, moisture content, feed composition, extruder type, screw configuration, barrel temperature, feed rate, and screw speed [3]. Furthermore, extrusion temperature is an essential component influencing extruder quality. Allai et al. [15] propose that adjusting the extrusion temperature can regulate several functional aspects of extrudates. As a result, research is required to quantify rice bran’s storage properties and estimate its shelf life using oxidation data to manage the quality of rice bran under specified storage circumstances and optimize its industrial application.

Therefore, the effects of extrusion treatment and temperature on storage stability, lipid oxidation, peroxidase (POD), peroxide values (PV), free fatty acids (FFA), fatty acid composition, and protein variations of RB and to provide an empirical and theoretical basis for maintaining the storage quality and elongating RB storage duration were studied. Since studies are frequently available on the application of extrusion treatment regarding extraction rate and the quality of RB oil, the information on the lipid oxidation and protein modifications of extruded RB during storage are yet to be fully explored and hence taken in hand the current study. 

## 2. Material and Methods

### 2.1. Material

Fresh rice bran (with moisture content (MC) of 21%) was purchased from Basu Rice Industry Co., Ltd., Yuanyang County, Xinxiang City, Henan Province. The RB was sieved (40 m aperture) to remove broken kernels, husk, and other external particles. 

### 2.2. Stabilization of RB

A self-cleaning extruder (SLG30-IV, Nanjing KY Chemical Machinery Co., Ltd., Nanjing, China) with twin-screw co-rotating and a screw diameter of 25 mm was used for the study. The machine has a barrel temperature of 120 °C, and a screw speed of 160 r/min was used to stabilize rice bran. The control group consisted of untreated RB.

### 2.3. Simulation of Storage Scenarios

The untreated RB and extruded RB samples were kept in desiccators with 70% relative humidity and stored at 15, 25, and 40 °C, respectively. Of rice bran samples, 3000 g of were kept in each storage group and analysed every ten days.

### 2.4. Experimental Design and Analysis

#### 2.4.1. Evaluation of Single-Factor Extrusion Treatment of RB

The screw speed, barrel temperature, and material moisture content were used to optimize the best extrusion condition. The residual peroxide activity (RPA) was used as an index for a single-factor test.

#### 2.4.2. Effect of Barrel Temperature on RPA

The RB was extruded at different barrel temperatures of 80, 100, 120, 140, and 160 °C while maintaining a fixed screw speed of 160 r/min and MC of 21%. The RPA corresponding to each single factor condition was determined. 

#### 2.4.3. Effect of Screw Speed on RPA

The RB was extruded at screw speeds of 120, 140, 160, 180, and 200 r/min while maintaining the barrel temperature and MC at 120 °C and 21%, respectively. The RPA corresponding to each single factor condition was determined. 

#### 2.4.4. Effect of Moisture Content on RPA

To evaluate the MC, the screw speed and barrel temperature were set to 160 r/min and 120 °C, respectively. The RB was then extruded and processed under treatment conditions of the material’s 15, 18, 21, 24, and 27% moisture content. The RPA corresponding to each single factor condition was determined.

### 2.5. Response Surface Methodology (RSM) Analysis

A response surface test with three factors and levels and Box–Behnken design with barrel temperature (A), screw speed (B), and material moisture content (C) as independent variables and peroxidase residual vitality (Y) as the dependent variable response values. The variable range was determined using information from the literature and our preliminary experiments. The factor level coding is shown in Table 1. The results were examined using a quadratic model to characterize the effect of factors, and the experimental data were fitted to the chosen model. The statistical significance of each response item was evaluated using the ANOVA (analysis of variance) method.

### 2.6. Quality Characteristics of Stabilized RB 

To determine the activity of peroxidase (POD), a mixture of bran enzyme extraction and substrate solution was prepared and incubated for 5 min at 37 °C. The absorbance was then measured at a wavelength of 420 nm [16]. Lipase activity was determined using p-nitrophenyl palmitate (pNPP) as the substrate. The reaction was incubated at 37 °C for 10 min, and the liberated p-nitrophenol was measured at 410 nm using a spectrophotometer (He et al., 2020). The free fatty acid (FFA) was determined using a standard titration. Of the sample, 1 g was centrifuged with 10 mL of ethanol and 50 mL of diethyl ether. The supernatant was titrated with 0.1 N HCl using phenolphthalein as an indicator and expressed as mg/100 g [17]. The peroxide values (PV) were determined per official AOAC methods and expressed as mEqO2/kg [18]. RB’s malondialdehyde content (MDA) was determined using 1 g of millet powder mixed with 9 mL of absolute ethanol and centrifuged at 4000 rpm for 10 min. The MDA content in the resulting supernatant was quantified using an MDA determination kit obtained from the Nanjing Jiancheng Bioengineering Institute in China, and results are expressed as µmol/L [19].

### 2.7. Nutritional Properties of Stabilized RB

#### 2.7.1. Determination of Amino Acid Composition

A sample was collected with a protein concentration between 10 and 20 mg. The hydrolysis tube was filled with 10 mL of 6 mol/L HCL and 3–4 drops of phenol. The hydrolysis tube was linked to the vacuum pump’s suction pipe, and the nitrogen was flushed repeatedly before the tube was sealed. A sealed hydrolysis tube was placed in an electric blast thermostat and heated at 110 °C for 22 h. The cooled tube was then filtered into a 100 mL volumetric flask to maintain a constant volume. In a 10 mL centrifuge tube, 1 mL of the hydrolysis solution was dried under reduced pressure, and the dried substance was then dissolved in 1 mL of water and dried again under reduced pressure. The dried residues were dissolved with sodium citrate buffer, shaken and mixed, filtered through a 0.22 membrane into a sample bottle, and analysed with an S 433 Amino Acid Analyzer (Sykam Chromatography, Eresing, Germany).

#### 2.7.2. Determination of Crude Protein Content 

The protein content of rice bran was determined using the Kjeldahl method with a Kjeldahl nitrogen analyzer (Beijing ZNQ Technology Co., Ltd., Beijing, China). Accurately weighed rice bran samples (0.5–2 g) were digested in concentrated sulfuric acid and a catalyst. The nitrogen content was determined by titration with a standardized sodium hydroxide solution. The nitrogen-to-protein conversion factor of 5.95 is commonly used for rice [20].

#### 2.7.3. Fatty Acid Composition (FAC) 

FACs of rice bran were determined using gas chromatography (GC) (Agilent 7890A, Santa Clara, CA, USA), furnished with a flame ionization detector [21]. Specific fatty acid was assessed by comparing their retention time (RT) to the prepared standard externally (Superco 37 component FAME mixture). The results were presented in relative area% [22].

### 2.8. Protein Electrophoresis Analysis of RB

The RB protein samples were dissolved in phosphate buffer at 25 °C for 2 h. The insoluble material was centrifugation at 7000 r/min for 30 min to remove them. The liquid remaining after centrifugation was combined in equal parts with a buffer solution consisting of 125 mmol/L Tris, 4.0% SDS, 30% glycerol, 2% β-mercaptoethanol, pH 6.8, and 0.002% bromophenol blue. The buffer used for the non-reducing samples did not contain 2% β-mercaptoethanol. The mixture was heated for 5 min, and 10 μL was used for each sample. The gel electrophoresis was performed using a 5% concentrated gel and a 12% resolving gel. The standard protein molecular weight (MW) range was 14.4–97.4 kDa. The gels were removed and fixed for 40 min, stained for 1 h, and decolourized until the background was transparent [23].

### 2.9. Statistical Analysis

The experiments were conducted thrice, and recorded the results were then expressed as the average value and standard deviation. The data were visualized using Origin 2022 software. Statistical analysis, including ANOVA and Duncan’s multiple comparisons, was carried out using IBM SPSS version 26.0, with a significance level of *p* < 0.05.

## 3. Results and Discussion

### 3.1. Single-Factor Results of Extrusion Treatment 

Different single factors demonstrated varying degrees of influence on POD in the single-factor test of rice bran extrusion treatment. The graphs revealed a polynomial relationship, and the ANOVA of the interaction demonstrates that the variables significantly affected the enzyme activity at *p* < 0.05. POD was significantly influenced by temperature, and higher temperatures led to better peroxidase residual activity inhibition (as shown in Figure 1a). 

At temperatures greater than 80 °C, the residual peroxidase activity rapidly decreased and reached its lowest level at 120 °C. It was observed that at extrusion temperatures maximizing at 128 °C, the enzyme was inactivated entirely regardless of the amount of water present [24], while POD did not change significantly above 120 °C. The POD decreased as the twin-screw rotational speed increased; after gradually increasing and stabilizing, the lowest POD reached 130 r/min (Figure 1b). Because of the short extruder holding time, the impact on the product was minimal [25]. Increasing the moisture in the rice bran might lower the discharge extrusion temperature, resulting in less inactivation. As per Figure 1c, the effect of MC on peroxidase residual activity showed a decreasing trend and then increased. The rate of reducing to the lowest POD value with rising MC in the low moisture range was faster than the rate of increase from the lowest value with increasing moisture content in the high. Increasing moisture did not influence enzyme inactivation due to a decrease in extrusion temperature caused by reduced heat generated through friction. The mechanism behind this occurrence can be explained by reducing the thickness or resistance to flow (viscosity) at higher moisture content. This allows heat dissipation generated by friction through mechanical energy [24]. The residual peroxidase activity was low at 17–23% MC and reached its lowest at 21% moisture content.

#### 3.1.1. RSM of Rice Bran by Extrusion

A single-factor experiment was conducted to identify the key factors affecting the process and determined that the barrel temperature, screw speed, and material moisture content were essential variables to investigate. A response surface experiment was conducted using the Box–Behnken principle (as outlined in Table 2) to analyse their impact further. 

The data collected were analysed using Design-Expert (version 10.0) software, and multiple regression equations were generated (as represented in Equation (1)).
(1)$$Y=2.64-0.50X_{1}-1.22X_{3}+0.84X_{1}X_{3}+1.12X_{2}X_{3}$$

The ANOVA Appendix A shows that the regression model is significant, and the lack of fit is not (*p* = 0.0779), indicating that the model has a good fit and high reliability. In addition, the F test can reflect the contribution rate of each factor. The larger the F value, the greater the impact on the test results. The most critical factors that influenced residual peroxidase activity were moisture content of the material > barrel temperature > and screw speed. The interactions between barrel temperature (*X*_1_) and material moisture content (*X*_3_), as well as between barrel temperature (*X*_1_), screw speed (*X*_2_), and material moisture content (*X*_3_), significantly affected residual peroxidase activity. However, neither screw speed (*X*_2_) nor the interaction between barrel temperature (*X*_1_) and screw speed (*X*_2_) had a significant impact. Furthermore, there was no discernible interaction in the residual peroxidase activity between barrel temperature, screw speed, or both.

#### 3.1.2. Optimization

Optimization is the processing condition of providing the optimal value of some functions that decide variables under specified limitations. As peroxidase activity is the most critical metric in determining enzyme inactivation satisfaction, the ideal process settings were chosen based on these characteristics. The software program utilized the optimization procedure to locate multiple relevant solutions, and the lowest peroxidase activity was selected as the appropriate condition. The minimum peroxidase residual activity conditions were calculated: barrel temperature 120 °C, screw speed 131 r/min, and material moisture content 21%. The optimum extrusion treatment conditions were set at a barrel temperature of 120 °C, a screw speed of 130 r/min, and a material MC of 21%. POD values were around 2.3% under these conditions. The relative error between the numerical model equations predicting POD in extruded stabilized rice bran and the measured values was small and within the acceptable range, indicating that the response surface optimization conditions were reliable.

### 3.2. Peroxidase Activity of Extruded RB during Storage

Figure 2 shows that after being subjected to extrusion conditions, RB’s residual enzymatic peroxidase activity gradually increases with increased storage time. 

This is due to the stabilized rice bran absorbing water again during storage to activate some peroxidase. Water serves as the heat transfer medium for extrusion, allowing the rice bran to be evenly heated and making it easier to control the residual activity of peroxidase [26]. The storage stability of peroxidase in extruded rice bran is significantly affected by the reabsorption of water. The extruded rice bran showed an increase in peroxidase activity during storage but remained below 5% at the end of the storage period. This suggests that the stabilization treatments applied to the rice bran effectively limit peroxidase activity. Additionally, it can be inferred that extruded rice bran exhibits superior storage stability compared to untreated rice bran. This also demonstrates that the residual peroxidase activity in rice bran is suitable for long-term storage when extrusion processing conditions are 130 °C, 130 r/min, and 11% water content. The quality of rice bran during storage is greatly affected by temperature. This is because temperature changes can alter enzyme activity, leading to an increase in oxidative rancidity. As shown in Figure 2, a longer storage time at a higher temperature results in a faster increase in residual peroxidase activity in rice bran. However, during the storage period, the gradual increase in POD at 15 and 25 °C was greater than at 40 °C. Ling et al. [27] discovered an increase in residual enzyme activity was observed in both RB and control samples that were treated with radiofrequency after 60 days of storage. The consistent rise in POD suggests that lipase and peroxidase have been inactivated and that no free radicals are present that could lead to further degradation [28].

### 3.3. Lipase Activity of Extruded RB during Storage

Rice bran storage stability can be evaluated by measuring lipase enzyme activity that plays a crucial role in preserving rice bran and catalyzing its oxidative rancidity. As shown in Figure 3, the lipase activity in rice bran can change over time, specifically after 60 days of storage at varying temperatures. 

During storage, extrusion-treated RB had approximately 20 times lower lipase activity than untreated RB. The findings revealed that the influence of storage conditions on enzyme activity was most significant at the start of storage. It is worth mentioning that the decrease in enzyme activities during storage could be due to hydroboration, as there was also a positive relationship found between enzyme activities and water activities in the wheat germ [29]. It is observed from the figure that the LPS activity of the treated rice bran increases with the increase of time during the storage period. The disintegration of the spherical membrane by protein hydrolysis, which increases the interaction between oil and lipase via diffusion, could cause increased lipase activity [30]. However, the LPS activity of the rice bran stored for 60 days was still lower than that of the untreated rice bran. Yan et al. [2] also found that infrared rice bran treatment had a limited impact on some inactivated lipase that regained activity during storage. This suggests that extrusion can effectively decrease rice bran lipase activity and preserve the stability of extruded rice bran during storage. Additionally, the study found that storage temperature did not significantly affect lipase activity in rice bran during storage, likely due to the strong inhibitory effect of extrusion treatment (*p* < 0.05). Similarly, Gong et al. [31] also reported that heating temperature does not significantly impact LPS activity during storage. However, the temperature increase could not enhance some lipases’ activity. This is also because lipase has poor thermal stability and temperature sensitivity. Higher temperature, pressure, and shear force during extrusion and puffing strongly inhibit it.

### 3.4. Free Fatty Acid (FFA) of Extruded RB during Storage

RB storage stability and quality are closely related to its fatty acid content and temperature. It was found that the FFA content of rice bran increased with the increase of temperature at different storage temperatures [32]. The content of free fatty acids in brown rice has no significant change under low-temperature storage [33]. The fatty acid content of extruded and untreated rice bran samples was observed over 60 days at different temperatures in Figure 4. The untreated rice bran samples showed a significant increase in fatty acid content (*p* < 0.05) as the initial value of 6.23 mg/100 g rose to 132.19 mg/100 g, 151.88 mg/100 g, and 156.61 mg/100 g after 60 days of storage at 15 °C, 25 °C, and 35 °C, respectively. However, the fatty acid content of extruded rice bran did not change significantly over the 60-day storage period at 15 °C. At 25 °C and 40 °C, the fatty acid content was 19.58 g/100 g after 50 days, with no significant change observed at the end of 60 days. Because of the abundance of enzymes in rice bran, the lipid composition is unstable during storage, resulting in a large amount of FFA and an increase in fatty acid value. Similarly, as storage temperature increased, so did the rate of increase in fatty acid value of rice bran, which was significantly (*p* < 0.05) associated with enzyme activity [34]. The fatty acid values of extrusion-treated rice bran (RB) during storage are lower than those of untreated RB stored at the same temperature. This can be seen in Figure 4, where at storage temperatures of 15 °C, 25 °C, and 40 °C, the fatty acid values of untreated rice bran are approximately 7, 8, and 9 times greater than those of extrusion-treated RB, respectively.

Despite a slight increase in fatty acid values at the end of storage at 25 °C and 40 °C, extrusion treatment still resulted in lower values overall. This pattern aligns with previous research on the subject [35], as an increase in free fatty acid (FFA) concentration in rice bran without extrusion heat treatment is typical. Raw rice bran is unsuitable for human food formulation due to the rapid production of FFA caused by hydrolysis and rancidity [36]. This demonstrates that extruded RB with a low fatty acid content has good storage quality. The fatty acid values of extruded RB increased only after 50 days of storage, indicating that extruded rice bran has a better stabilization effect. Furthermore, as the storage temperature dropped, the lipase activities of RB without extrusion processing decreased.

### 3.5. Peroxide Values of Extrusion RB during Storage

The peroxide value can characterize the degree of fat oxidation to a certain extent. Various external factors decompose the fat in rice bran during storage, producing many free fatty acids. The intermediate products’ FFA is unstable and could be further oxidized into peroxides, destroying the stability and quality of rice bran. Figure 5 depicts the change in peroxide value of extruded and untreated RB samples stored for 60 days at 15 °C, 25 °C, and 40 °C. The peroxide value of untreated rice bran increased significantly at different storage temperatures, as seen in Figure 5. 

The initial peroxide value of untreated RB was 1.71 mEqO2/kg, but it rose to 2.86, 2.89, and 3.15 mEqO2/kg after 60 days of storage at 15, 25, and 40 °C, respectively. On the other hand, extruded rice bran had an initial peroxide value of 2.35 mEqO2/kg, which increased to 2.78, 2.83, and 3.04 mEqO2/kg after storage at 15, 25, and 40 °C, respectively. It was observed that the peroxide values of rice bran increased with storage time at each temperature, and the higher the temperature, the higher the peroxide value.

The gradual increase in PV implies that lipase and peroxidase have been inactivated and that no free radicals are present to cause further degradation [28]. As a result, peroxide formation is inhibited. This conformed with the findings of Talcott et al. [37], who discovered that the peroxide content of peanuts stored at 25 °C for 4 months was half that of peanuts stored at 35 °C for the same period. On the other hand, [38] discovered that the peroxide value of pine nuts during storage was positively correlated with storage temperature and time. Because of the influence of time and temperature during the treatment process, the rice bran had varying degrees of oxidative rancidity, increasing its peroxide value. The peroxide values of extruded rice bran were lower at the end of storage than at all temperatures for untreated rice bran [39]. This suggests that the stability of stabilized rice bran had improved. The changes in RB peroxide could be attributed to the initial rice bran value before treatment, whereas lipase was affected by temperature and exposure period [6]. The rice bran can vary in oxidation and rancidity depending on the temperature and duration of treatment, leading to an increase in its peroxide value. However, using extrusion methods can improve the stability of the rice bran and lower its peroxide value compared to untreated rice bran. After two months of storage under various conditions, the peroxide values of the rice bran samples were found to be below the permissible limit of 10 mEqO2/kg established by the Codex Alimentarius Commission. These findings are consistent with those obtained when RB was stabilized using a microwave [40].

### 3.6. Malondialdehyde Content of Extruded RB during Storage

Malondialdehyde is an intermediate product of lipid membrane peroxidation that can aggravate cell membrane damage and reflect the oxidation level of rice bran. In various studies, malondialdehyde has also been linked to grain and oil crop storage stability [41]. The data in Figure 6 illustrates the variations in the malondialdehyde present in extruded rice bran and untreated rice bran that were stored at 15, 25, and 40 °C for 60 days. 

It was observed that the malondialdehyde content of the samples in each storage group significantly increased (*p* < 0.05) as the storage temperature and duration were elevated. Additionally, the research found that the activity of enzymes in rice bran decreased with prolonged storage time, which slowed down the production of malondialdehyde.

Additionally, continuous high temperatures caused some malondialdehyde to evaporate, decreasing the amount of malondialdehyde. During storage, MDA content changed as well. The higher the temperature, the greater the rice bran’s oxidation and malondialdehyde content [42]. This rising trend of malondialdehyde content is in line with the study of Mexis and Kontominas [43], in which the malondialdehyde content of walnuts stored at low temperatures was lower than that of walnuts stored at high temperatures. Free fatty acids accumulate significantly during the accelerated storage, and hydrogen peroxide is produced due to lipoxygenase activation and fat oxidation [44]. As its decomposition product, malondialdehyde starts accumulating, increasing the malondialdehyde level. The data in Figure 6 show a correlation between the level of fatty acids and changes in the content of MDA, which is a natural result of MDA’s role as an intermediate in the oxidation of lipids. However, it was found that the initial MDA content in stabilized rice bran was higher than in untreated rice bran. Extrusion treatment improved the stability and quality of stabilized rice bran, as evidenced by the decrease in MDA concentration, fatty acid, and peroxide values.

### 3.7. Fatty Acid Composition of Extruded RB during Storage

Assessing fatty acid value is crucial in determining rice bran’s storage stability and quality. The changes in the fatty acid composition and content can be used to evaluate the storage stability of rice bran and its quality to some extent. As can be seen in Table 3, the fatty acid composition of rice bran changes during storage under different temperature conditions. Rice bran is primarily composed of unsaturated fatty acids, with high levels of oleic and linoleic acid, accounting for nearly 80% of the total. The fatty acid concentration found in this study is comparable with that revealed by several other studies [22,45]. 

Unsaturated fatty acids account for more than 80% of total fatty acids in RB, with oleic acid dominating (40.40%), followed by linoleic acid (38.95%) and g-linolenic acid (0.46%). While palmitic acid (16.79%), arachidic acid (1.57%), and stearic acid (1.20%) were identified as significant saturated fatty acids. The data obtained indicate similar findings as Yilmaz et al. [45] that the extrusion impact in RB does not substantially modify the fatty acid concentration (*p* < 0.05). Likewise, Li et al. [29] discovered that infrared heating did not significantly diminish the primary fatty acids in wheat germ. The research found that various types of fatty acids had distinct reactions to varying storage temperatures. Additionally, it was determined that the changes in fatty acid composition in rice bran after undergoing extrusion treatment were minimal. On the one hand, because acidification and oxidation of rice bran is a multiple-material dynamic change process, rice bran storage stability improved after extrusion treatment. The fatty acids in extruded rice bran changed significantly (*p* < 0.05) but remained relatively stable. The oleic acid content in untreated rice bran decreased while linoleic acid increased as storage temperature and time increased. This result could be due to the oxidative decomposition of fatty acids caused by high temperatures; the decomposition rate of oleic acid is more significant than its generation rate, whereas the increase in linoleic acid content could be due to its generation being more significant than its decomposition rate. Because oxidative decomposition is a dynamic process, the expansion of total saturated fatty acid content, decreased oleic acid content, and increased linoleic acid content in untreated rice bran cannot directly determine the degree of oxidation. The stability performance of extruded rice bran can be evaluated using the ratio of oleic acid to linoleic acid (O/L). This measurement can indicate storage stability to a certain extent. In this study, the O/L values of all samples were found to not change significantly during the storage period, indicating that the extruded rice bran was relatively stable during storage.

### 3.8. Crude Protein Treated by Extruded RB during Storage

Table 4 illustrates the variations in the crude protein content of rice bran during storage at different temperatures. 

The data indicate that the crude protein content of rice bran under extrusion treatment does not show a significant difference (*p* < 0.05) compared to the control group. These findings align with Ertaş [46], who found 14.2% crude protein in rice by-products. Additionally, our results show that the extrusion treatment does not have a significant effect (*p* < 0.05) on the crude protein content of rice bran. These findings align with previous studies conducted by Khan et al. [47], Oliveira et al. [48], and Capellini et al. [49] which also found that the crude protein content of rice bran remains relatively unchanged as storage temperature and period increases. The increase in protein content may be attributed to the rice bran’s high protein and fat content.

### 3.9. Amino Acid Profile of Extruded RB during Storage

The nutritional value of rice bran is significantly impacted by the changes that occur during its storage period. The composition and content of amino acids are the final hydrolyzed products of RB protein. As seen in Table 5 and Table 6, there were significant differences in the amino acid composition and content of rice bran when stored at different temperatures. 

However, no clear trend was observed in the changes of amino acids across different storage temperatures. Glutamic acid was the most abundant amino acid in untreated (10.17 mg/g) and extruded (4.61 mg/g) RB at various temperatures throughout the storage period. Cystine was the least abundant in untreated (0.22 mg/g) and extruded (0.11 mg/g) RB. Similar results were found by Akinyede et al. [50] in cereal breakfasts of millet and rice bran blends. The alteration in the amino acid composition of rice bran protein concentrate can be attributed to microbial activity and development, leading to changes in protein breakdown and the formation of new proteins [51]. As storage temperature and duration increase, aspartic acid, valine, and proline levels in untreated rice bran also increase significantly. Likewise, Kim et al. [52] revealed that total amino acid in RB increased 3–5 times by the anaerobic storage conditions. The significance level (*p* < 0.05) of amino acids increases with the storage temperature. Most of the amino acid contents did not change significantly (*p* > 0.05) during the storage period, except the cysteine and methionine contents, which were relatively low due to the influence of acid hydrolysis. Protein enzymatic degradation may increase the total amino acid content. The study found that as storage time and temperature increased, most amino acids decreased in extruded treated rice bran compared to untreated samples. This is likely due to the extrusion process involving high pressure, shear, friction, and temperature. These conditions can damage the protein’s structure, break intermolecular bonds, and cause protein denaturation. Additionally, the extrusion process can also lead to the production of amino acids by cracking some of the protein [13,53]. One possibility behind this is that the internal structure of rice bran has been changed by extrusion.

### 3.10. Protein Electrophoresis Analysis Extruded RB during Storage

The impacts of extrusion treatment on oxidative protein stability were investigated by evaluating changes in its molecular weight using electrophoresis during storage. Figure 7 depicts the electrophoretic analysis of rice bran protein under different extrusion treatments: a and b at 15 °C, c and d at 25 °C, and e and f at 40 °C storage samples represent the reduced and non-reduced electrophoretic analysis of untreated rice bran; g and h at 15 °C, i and j at 25 °C, k and l at 40 °C storage samples refer to reduced and non-reduced electrophoresis analysis of extruded rice bran. Figure 7 shows the primary molecular weights of rice bran protein bands distribution around 64 kD, 38 kD, and 24 kD. Such protein bands become less apparent as they widen during storage. 

This finding demonstrates that protein oxidation may generate various products with varying molecular weights [54]. Rice bran subunits have molecular weights ranging from 14 to 66 kDa, consistent with Ling et al. [55] findings. The researchers found that the molecular weight of the RB protein subunits ranged from 10 to 55 kDa. Analysis of the non-reducing electropherogram revealed that the critical component of rice bran is highly stable and resistant to temperature and storage time changes during extrusion. No significant variations were detected throughout the entire storage period. The protein band pattern is similar to the rice protein composition described by Watanabe et al. [56]. As a result, protein concentrates such as glutenin, gliadin, albumin, and globulin may be contained in the original rice bran protein [57]. However, non-reduced electrophoresis reveals that the extruded rice bran protein tends to deepen at the large molecular weight of the bands at the end of storage, which could be caused by protein molecule generation or polymerization.

### 3.11. Degradation Kinetics Analysis

The degradation kinetics model accurately predicted quality changes during food preparation and storage (Table 7). In this study, we utilized the Arrhenius equation to conduct a dynamic analysis of the characteristics of rice bran. 

The Arrhenius equation is a widely used model to evaluate the effect of temperature on physicochemical reactions and process rates in food systems [58]. We modelled the decrease in enzyme activity over time during a two-month storage period using three different kinetic models: zero-order, first-order, and second-order. We used the coefficient of determination (R^2^) and the rate constant (k) obtained from the regression equation to evaluate the accuracy of each model and determine linearity (Appendix A). The rate of three enzymatic reactions, peroxidase inactivation, peroxide activity, and changes in fatty acid levels in stabilized rice bran during storage, was analysed using different kinetic models, including zero-order, first-order, and second-order. The results showed that the first-order kinetic model had a higher correlation coefficient (R^2^) for each parameter than the zero-order kinetic model. In general, when R^2^ is higher, it indicates that the mathematical model matches the actual data and that there is little to no variation between the predicted value and the observed value [41]. The degradation kinetics of peroxidase, peroxide, and lipase activities in rice bran samples best fit the first-order models, with the R^2^ ranging from 0.868 to 0.996. 

At the same time, fatty acid values have lower values ranging from 0.639 to 0.935. The chemical and physical changes during food processing and storage often follow the pattern of first-order reaction kinetics [59], consistent with our findings that rice bran qualities that correspond to the Arrhenius diagram change with temperature while others do not. The lowest rate constants (k) were found in second-order kinetics ranging from 0.0065 to 0.00849 for peroxidase, −0.00103 to −0.0042 for peroxide value, 1.88 × 10^−4^ to −3.46 × 10^−4^ for lipase activity and −7.53 × 10^−6^ to −1.68 × 10^−3^ for fatty acid values. Rice bran showed the lowest k, R^2^, RMSE, and X^2^ for fatty acid values in second-order kinetics compared to zero- and first-order kinetics. Enzymatic stabilization of rice bran leads to lower k, RMSE, and X^2^ values for peroxide, lipase activity, and fatty acid content than untreated stabilized rice bran. Additionally, it has been observed that enzymatic stabilization results in a higher rate of degradation of free fatty acids, as they oxidize more quickly than triacylglycerols. A significant number of free radicals in oxidized lipids causes the breakdown of antioxidant substances. Lipase hydrolysis generates free fatty acids, which are easily oxidized by lipoxygenase. This leads to the depletion of antioxidants and vitamin E due to the antioxidant effect [60]. As a result, rice bran’s stability may limit the enzyme destruction rate. Consequently, storage at a low temperature retards the deterioration of physical and chemical features, preserving the high quality of rice bran. Hence, our kinetic investigation based on the Arrhenius model can better explain temperature-dependent variations. The kinetic analysis demonstrated that the enzyme activity increased as storage duration increased, whereas the fatty acids did not change much. Therefore, our findings may aid in developing a preservation method that can retain the quality of rice bran for ultimate usage.

## 4. Conclusions

This study investigated the effects of extrusion treatment on the storage stability and nutrient composition of rice bran (RB), including changes in enzyme activity, fatty acid composition, and protein nutritional value. The findings demonstrate that extrusion treatment can improve the storage stability of RB and help maintain its nutrient composition by reducing POD and LPS activities. However, prolonged storage time can lead to an increase in lipid oxidation and rancidity. Moreover, while the fatty acid composition remains stable during storage, the total number of amino acids tends to decrease, resulting in a declining trend for both E/T and E/N. The study also provides insights into the degradation of enzymes and protein subunits in RB during storage, which can be effectively modelled using first-order kinetics. The study also discovered that the degradation of enzymes such as peroxidase, peroxide, and lipase in RB samples could be explained using first-order models with high R^2^ values between 0.868 and 0.996. Overall, these findings are essential for enhancing the utilization of RB as a valuable source of nutrients and functional ingredients in various food applications. Further research is warranted to explore strategies for improving RB’s protein nutritional value and stability during storage.

## Figures and Tables

**Figure 1 foods-12-01236-f001:**
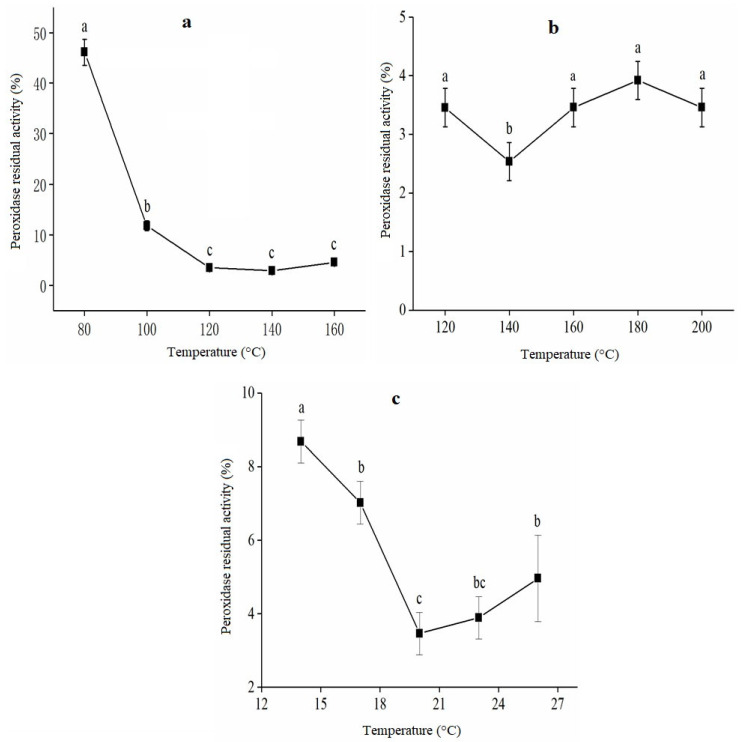
Effect of barrel temperature (**a**), screw speed material (**b**), and moisture content (**c**) on residual activity of peroxidase in extruded rice bran during storage. The letters in the figure indicate significant differences between different treatment times and storage conditions, with a significance level of *p* < 0.05.

**Figure 2 foods-12-01236-f002:**
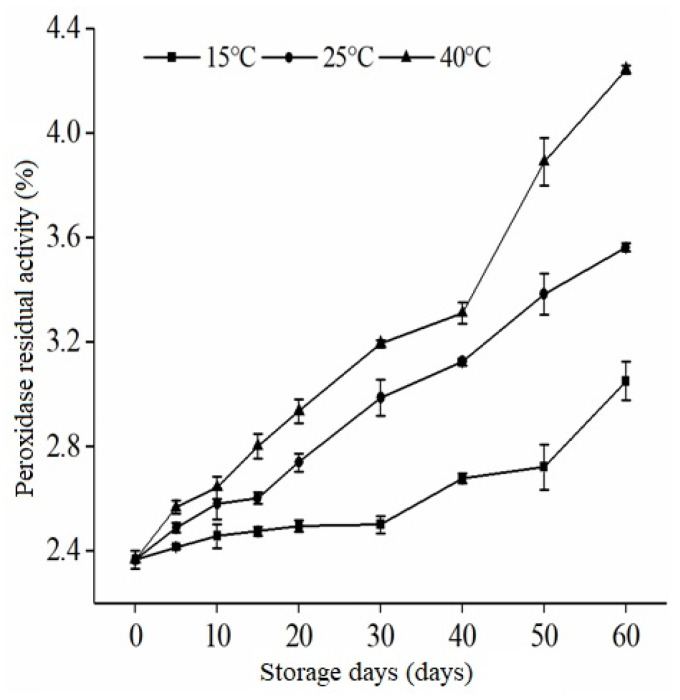
Effect of Storage Temperature on Residual Peroxidase Activity in Extruded Rice Bran.

**Figure 3 foods-12-01236-f003:**
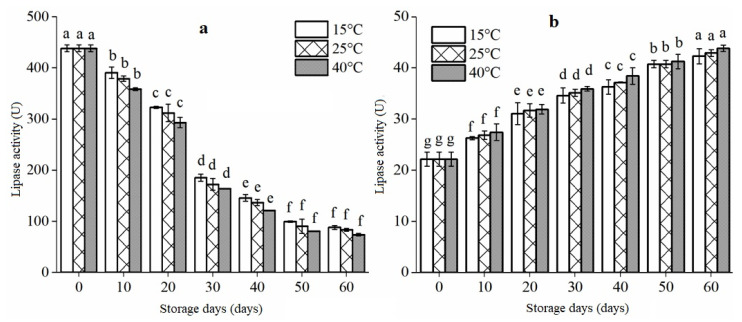
Changes in lipase activity of rice bran treated by extrusion. Note: (**a**) untreated rice bran; (**b**) extruded rice bran. The letters in the figure indicate significant differences between different treatment times and storage conditions, with a significance level of *p* < 0.05.

**Figure 4 foods-12-01236-f004:**
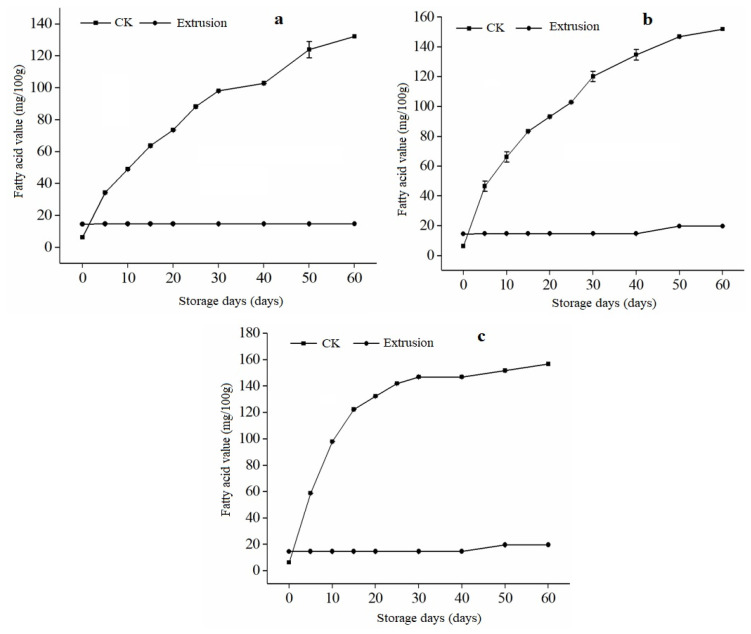
Changes in the fatty acid value of rice bran treated by extrusion. Note; (**a**–**c**) in the figure refer to the storage temperature of 15, 25, and 40 °C; CK is untreated rice bran.

**Figure 5 foods-12-01236-f005:**
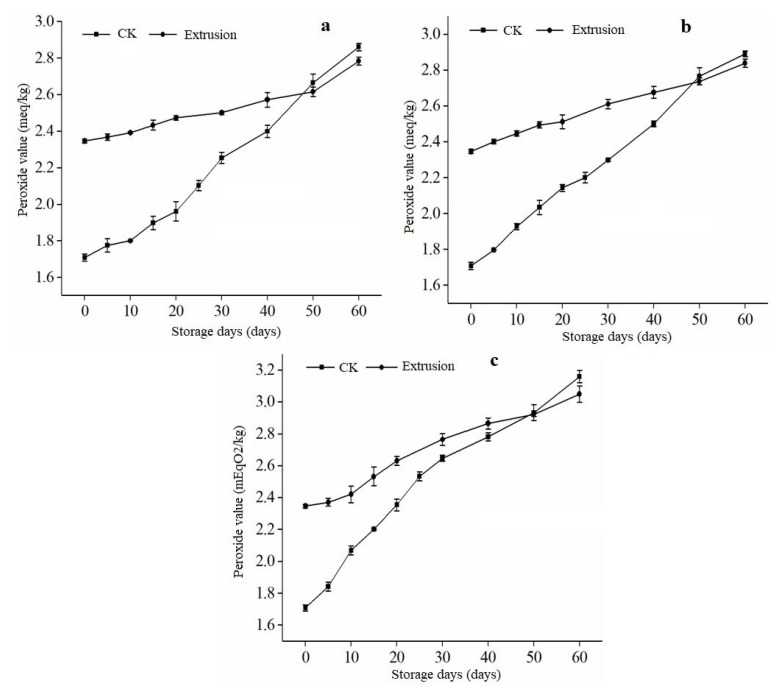
Changes in peroxide value of rice bran treated by extrusion. Note; (**a**–**c**) in the figure refer to the storage temperature of 15 °C, 25 °C, and 40 °C; CK is untreated rice bran.

**Figure 6 foods-12-01236-f006:**
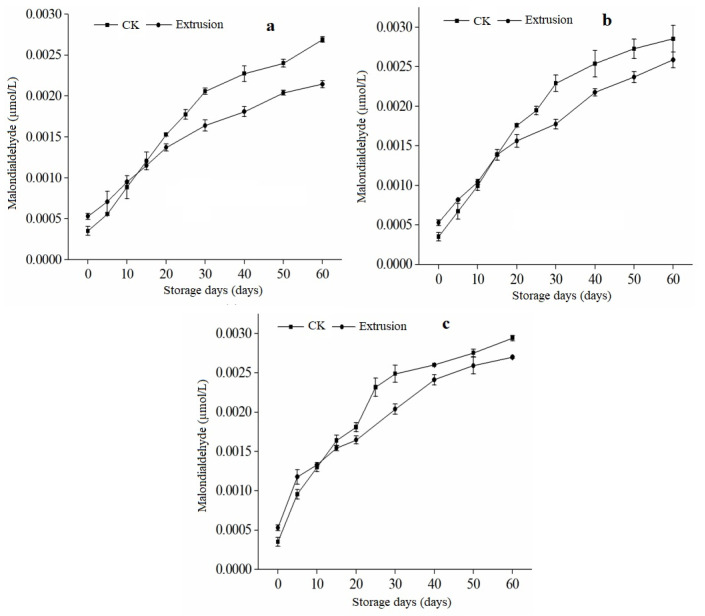
Changes in malondialdehyde content of rice bran treated by extrusion. Note; (**a**–**c**) in the figure refer to the storage temperature of 15 °C, 25 °C, and 40 °C; CK is untreated rice bran.

**Figure 7 foods-12-01236-f007:**
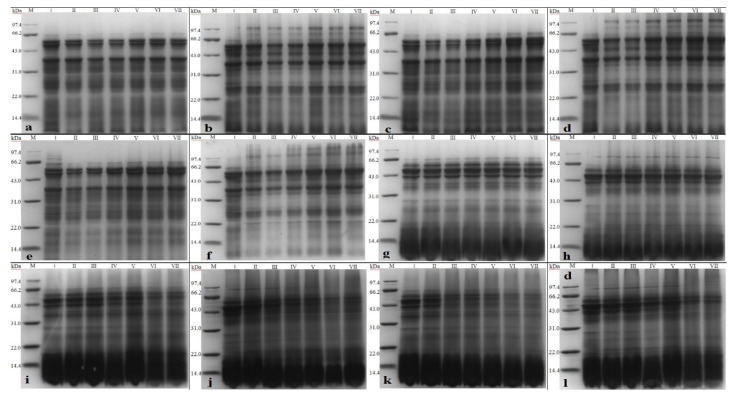
Electrophoresis analysis of extruded rice bran protein. Note: (**a**,**b**) in the figure represent the protein non-reducing and reducing electrophoresis of untreated rice bran stored at 15 °C, respectively; (**c**,**d**) represent the non-reducing and reductive electrophoresis of untreated rice bran stored at 25 °C, respectively. (**e**,**f**) non-reducing protein and reducing electrophoresis of untreated rice bran stored at 40 °C; (**g**,**h**), respectively, are the non-reducing and reductive electrophoresis of extruded rice bran stored at 15 °C.; (**i**,**j**) represent the protein non-reducing and reduction electrophoresis of the extruded rice bran stored at 25 °C, respectively; (**k**,**l**) represent the non-reduced protein of the extruded rice bran stored at 40 °C, respectively. Prototype electrophoresis and reduction electrophoresis analysis; lanes I–VII are protein non-reduced and reduced electrophoresis analysis of rice bran stored at 0, 10, 20, 30, 40, 50, and 60 d, respectively.

**Table 1 foods-12-01236-t001:** Factor level coding table of single-factor extrusion treatment of RB.

Factors	Code		Level		
	Code	Non-Coding	1	0	−1
Barrel temperature (°C)	*X* _1_	A	140	120	100
Screw speed (r/min)	*X* _2_	B	160	140	120
Moisture (%)	*X* _3_	C	23	20	17

**Table 2 foods-12-01236-t002:** Response surface optimization (Box–Behnken design) of residual peroxidase activity.

Test Number	Factor Level			POD Residual Vitality (%)
	Barrel Temperature (*X*_1_) (°C)	Screw Speed (*X*_2_) (r/min)	Material Moisture Content (*X*_3_) (%)	
1	0	0	0	2.28
2	1	0	1	4.42
3	1	1	0	3.46
4	0	0	0	3.12
5	0	−1	1	3.26
6	0	0	0	2.60
7	−1	1	0	5.73
8	0	0	0	2.50
9	1	0	−1	5.76
10	−1	0	1	3.11
11	−1	−1	0	4.31
12	1	−1	0	3.38
13	−1	0	−1	7.83
14	0	1	−1	6.07
15	0	−1	−1	7.37
16	0	1	1	6.45
17	0	0	0	2.72

**Table 3 foods-12-01236-t003:** Changes in the fatty acid composition in untreated and extruded rice bran during storage.

Storage Temp (°C)	Storage Days (d)	Fatty Acid Composition (%)										
		Myristic Acid	Palmitic Acid	Stearic Acid	Oleic Acid	Linoleic Acid	Linolenic Acid	Arachidic Acid	Arachidonic Acid	Saturated Fatty Acid	Unsaturated Fatty Acids	O/L
Untreated (°C) samples												
15 °C	0	0.17 ± 0.00 ^a^	16.64 ± 0.02 ^a^	1.19 ± 0.14 ^a^	40.81 ± 0.26 ^a^	38.61± 0.00 ^b^	0.48 ± 0.04 ^a^	1.64 ± 0.01 ^a^	0.45 ± 0.16 ^a^	19.65 ± 0.14 ^a^	80.35 ± 0.14 ^b^	1.06 ± 0.01 ^a^
	60	0.19± 0.03 ^aA^	15.13 ± 0.23 ^bA^	1.19 ± 0.00 ^aA^	41.31 ± 0.21 ^aA^	39.55 ± 0.03 ^aA^	0.43 ± 0.01 ^aB^	1.66 ± 0.01 ^aA^	0.54 ± 0.01^B^	18.17 ± 0.25 ^bA^	81.83 ± 0.25 ^aA^	1.04 ± 0.00 ^aB^
25 °C	0	0.17 ± 0.00 ^a^	16.64 ± 0.02 ^a^	1.19 ± 0.14 ^a^	40.81 ± 0.26 ^a^	38.61± 0.00 ^b^	0.48 ± 0.04^a^	1.64 ± 0.01 ^a^	0.45 ± 0.16 ^a^	19.65 ± 0.14 ^a^	80.35 ± 0.14 ^b^	1.06 ± 0.01 ^a^
	60	0.20 ± 0.02 ^aA^	15.03 ± 0.11 ^bA^	1.21 ± 0.04 ^aA^	41.70 ± 0.22 ^aA^	39.15 ± 0.04 ^aB^	0.49 ± 0.01 ^aA^	1.66 ± 0.02 ^aA^	0.56 ± 0.01 ^aAB^	18.10 ± 0.19 ^bA^	81.90 ± 0.19 ^aA^	1.07 ± 0.01 ^aA^
40 °C	0	0.17 ± 0.00 ^a^	16.64 ± 0.02 ^a^	1.19 ± 0.14 ^a^	40.81 ± 0.26 ^a^	38.61± 0.00 ^b^	0.48 ± 0.04 ^a^	1.64 ± 0.01 ^a^	0.45 ± 0.16 ^a^	19.65 ± 0.14 ^a^	80.35 ± 0.14 ^b^	1.06 ± 0.01 ^a^
	60	0.17 ± 0.01 ^aA^	14.89 ± 0.02 ^bA^	1.16 ± 0.01 ^aA^	41.76 ± 0.11 ^bA^	39.26 ± 0.07 ^aB^	0.52 ± 0.00 ^aA^	1.67 ± 0.00 ^aA^	0.57 ± 0.00 ^aA^	17.90 ± 0.04 ^bA^	82.10 ± 0.04 ^bA^	1.06 ± 0.00 ^aA^
Extrusion (°C) treated samples												
15 °C	0	0.19 ± 0.00 ^a^	16.51 ± 0.12 ^a^	1.22 ± 0.01 ^a^	40.38 ± 0.07 ^a^	39.04 ± 0.18 ^a^	0.52 ± 0.02 ^a^	1.61 ± 0.02 ^a^	0.53 ± 0.00 ^a^	19.52 ± 0.12 ^a^	80.48 ± 0.12 ^a^	1.03 ± 0.01 ^a^
	60	0.19 ± 0.03 ^aA^	16.79 ± 0.19 ^aA^	1.20 ± 0.11 ^aA^	40.40 ± 0.46 ^aA^	38.95 ± 0.49 ^bA^	0.46 ± 0.04 ^aAB^	1.57 ± 0.01 ^aA^	0.44 ± 0.13 ^aA^	19.75 ± 0.07 ^aA^	80.25 ± 0.07 ^aB^	1.04 ± 0.02 ^aA^
25 °C	0	0.19 ± 0.00 ^a^	16.51 ± 0.12 ^a^	1.22 ± 0.01 ^a^	40.38 ± 0.07 ^a^	39.04 ± 0.18 ^a^	0.52 ± 0.02 ^a^	1.61 ± 0.02 ^a^	0.53 ± 0.00 ^b^	19.52 ± 0.12 ^a^	80.48 ± 0.12 ^a^	1.03 ± 0.01 ^a^
	60	0.18 ± 0.01 ^aA^	16.33 ± 0.23 ^aA^	1.11 ± 0.10 ^aA^	40.68 ± 0.47 ^aA^	39.15 ± 0.32 ^aA^	0.40 ± 0.02 ^bB^	1.60 ± 0.01 ^aA^	0.55 ± 0.00 ^aA^	19.22 ± 0.12 ^aB^	80.78 ± 0.12 ^aA^	1.04 ± 0.02 ^aA^
40 °C	0	0.19 ± 0.00 ^a^	16.51 ± 0.12 ^a^	1.22 ± 0.01 ^a^	40.38 ± 0.07 ^a^	39.04 ± 0.18 ^a^	0.52 ± 0.02 ^a^	1.61 ± 0.02 ^a^	0.53 ± 0.00 ^a^	19.52 ± 0.12 ^a^	80.48 ± 0.12 ^a^	1.03 ± 0.01 ^a^
	60	0.18 ± 0.02 ^aA^	16.54 ± 0.01 ^aA^	1.32 ± 0.03 ^aA^	40.50 ± 0.04 ^aA^	38.84 ± 0.03 ^aA^	0.50 ± 0.00 ^aA^	1..58 ± 0.02 ^aA^	0.54 ± 0.02 ^aA^	19.62 ± 0.05 ^aA^	80.38 ± 0.05 ^aB^	1.04 ± 0.00 ^aA^

Lowercase letters indicate differences in storage periods at the same temperature and treatment condition at *p* < 0.05, while uppercase letters indicate differences in treatment times at the same temperature and storage period at *p* < 0.05.

**Table 4 foods-12-01236-t004:** Changes in crude protein content in untreated and extruded rice bran during storage.

Temperature (°C)	Crude Protein Content (%)						
	Storage Days (d)						
	0	10	20	30	40	50	60
Untreated (°C) samples							
15 °C	13.37 ± 0.12 a	13.43 ± 0.03 a	13.38 ± 0.03 a	13.28 ± 0.07 a	13.39 ± 0.07 a	13.38 ± 0.11 a	13.39 ± 0.09 a
25 °C	13.37 ± 0.12 a	13.47 ± 0.02 a	13.49 ± 0.06 a	13.56 ± 0.27 a	13.42 ± 0.08 a	13.51 ± 0.24 a	13.53 ± 0.21 a
40 °C	13.37 ± 0.12 a	13.26 ± 0.28 a	13.32 ± 0.09 a	13.32 ± 0.03 a	13.25 ± 0.23 a	13.34 ± 0.07 a	13.34 ± 0.17 a
Extrusion (°C) treated samples							
15 °C	13.37 ± 0.12 a	13.52 ± 0.38 a	13.48 ± 0.30 a	13.52 ± 0.19 a	13.37 ± 0.28 a	13.54 ± 0.31 a	14.49 ± 0.03 a
25 °C	13.55 ± 0.46 a	13.64 ± 0.20 a	13.55 ± 0.09 a	13.61 ± 0.24 a	13.20 ± 0.21 a	13.38 ± 0.25 a	13.49 ± 0.04 a
40 °C	13.47 ± 0.24 a	13.36 ± 0.14 a	13.46 ± 0.04 a	13.44 ± 0.16 a	13.51 ± 0.22 a	13.39 ± 0.20 a	13.41 ± 0.12 a

Lowercase letters indicate differences in storage periods at the same temperature and treatment condition at *p* < 0.05.

**Table 5 foods-12-01236-t005:** Amino acid composition and content changes in untreated rice bran during storage.

Amino Acid (mg/g)				Untreated Rice Bran					
	15 °C			25 °C			40 °C		
	0 Day	30 Days	60 Days	0 Day	30 Days	60 Days	0 Day	30 Days	60 Days
Aspartic acid	4.22 ± 0.03 b	4.35 ± 0.04 bB	4.53 ± 0.07 aA	4.22 ± 0.03 b	4.51 ± 0.01 bA	4.68 ± 0.07 aA	4.22 ± 0.03 b	4.58 ± 0.01 aA	4.71 ± 0.07 aA
Threonine	2.02 ± 0.11 a	2.02 ± 0.07 aA	2.11 ± 0.11 aA	2.02 ± 0.11 a	2.05 ± 0.04 aA	2.07 ± 0.11 aA	2.02 ± 0.11 a	2.06 ± 0.17 aA	2.00 ± 0.14 aA
Serine	2.34 ± 0.06 a	2.45 ± 0.07 aA	2.53 ± 0.07 aA	2.34 ± 0.06 a	2.54 ± 0.07 aA	2.60 ± 0.04 aA	2.34 ± 0.06 a	2.53 ± 0.07 aA	2.57 ± 0.11 aA
Glutamic acid	10.17 ± 0.03 a	10.16 ± 0.10 aA	10.49 ± 0.49 aA	10.17 ± 0.03 a	10.56 ± 0.68 aA	10.7 ±0.68 aA	10.17 ± 0.03 a	10.8 ±0.19 aA	10.96 ± 0.11 aA
Glycine	3.62 ± 0.03 a	3.67 ± 0.06 aA	3.64 ± 0.08 aA	3.62 ± 0.03 a	3.66 ± 0.01 aA	3.60 ± 0.09 aA	3.62 ± 0.03 a	3.63 ± 0.02 aA	3.65 ± 0.04 aA
Alanine	4.04 ± 0.20 a	4.06 ± 0.18 aA	4.06 ± 0.03 aA	4.04 ± 0.20 a	4.09 ± 0.07 aA	4.02 ± 0.03 aA	4.04 ± 0.20 a	4.05 ± 0.16 aA	4.01 ± 0.17 aA
Cystine	0.22 ± 0.01 a	0.24 ± 0.01 aA	0.24 ± 0.04 aA	0.22 ± 0.01 a	0.22 ± 0.03 aA	0.22 ± 0.01 aA	0.22 ± 0.01 a	0.26 ± 0.03 aA	0.23 ± 0.03 aA
Valine	3.22 ± 0.07 b	3.22 ± 0.04 bB	3.48 ± 0.10 aA	3.22 ± 0.07 b	3.54 ± 0.04 aA	3.56 ± 0.06 aA	3.22 ± 0.07 b	3.48 ± 0.03 aA	3.56 ± 0.04 aA
Methionine	1.11 ± 0.05 a	1.08 ± 0.14 aA	1.06 ± 0.12 aA	1.11 ± 0.05 a	1.16 ± 0.08 aA	1.09 ± 0.11 aA	1.11 ± 0.05 a	1.15 ± 0.02 aA	1.11 ± 0.11 aA
Isoleucine	3.00 ± 0.14 a	3.04 ± 0.04 aA	3.04 ± 0.03 aA	3.00 ± 0.14 a	3.08 ± 0.05 aA	3.07 ± 0.07 aA	3.00 ± 0.14 a	3.08 ± 0.02 aA	3.06 ± 0.08 aA
Leucine	6.08 ± 0.14 a	6.41 ± 0.13 aA	6.45 ± 0.19 aB	6.08 ± 0.14 a	6.53 ± 0.28 aA	6.64 ± 0.02 aAB	6.08 ± 0.14 a	6.68 ± 0.12 aA	6.94 ± 0.02 aA
Tyrosine	1.73 ± 0.02 a	1.75 ± 0.09 aA	1.75 ± 0.08 aA	1.73 ± 0.02 a	1.77 ± 0.03 aA	1.73 ± 0.04 aA	1.73 ± 0.02 a	1.74 ± 0.09 aA	1.75 ± 0.07 aA
Phenylalanine	2.07 ± 0.07 a	2.07 ± 0.09 aA	2.07 ± 0.18 aA	2.07 ± 0.07 a	2.05 ± 0.10 aA	2.04 ± 0.04 aA	2.07 ± 0.07 a	2.08 ± 0.00 aA	2.10 ± 0.12 aA
Histidine	2.65 ± 0.02 a	2.71 ± 0.07 aC	2.85 ± 0.17 aA	2.65 ± 0.02 a	3.09 ± 0.04 aA	3.11 ± 0.23 aA	2.65 ± 0.02 a	2.94 ± 0.01 aB	3.00 ± 0.09 aA
Lysine	4.31 ± 0.01 a	4.32 ± 0.13 aA	4.30 ± 0.12 aA	4.31 ± 0.01 a	4.29 ± 0.08 aA	4.30 ± 0.11 aA	4.31 ± 0.01 a	4.33 ± 0.03 aA	4.30 ± 0.06 aA
Arginine	7.80 ± 0.14 a	7.84 ± 0.25 aA	7.80 ± 0.16 aA	7.80 ± 0.14 a	7.87 ± 0.30 aA	7.87 ± 0.06 aA	7.80 ± 0.14 a	7.88 ± 0.38 aA	7.87 ± 0.06 aA
Proline	2.50 ± 0.04 b	2.51 ± 0.24 bB	3.25 ± 0.27 aA	2.50 ± 0.04 b	3.33 ± 0.15 aA	3.29 ± 0.19 aA	2.50 ± 0.04 b	3.20 ± 0.22 aA	3.24 ± 0.12 aA
TAA	61.12 ± 0.69 b	61.90 ± 0.16 a bB	63.64 ± 0.81 aA	61.12 ± 0.69 b	64.36 ± 0.21 aA	64.61 ± 0.11 aA	61.12 ± 0.69 b	64.54 ± 0.83 aA	65.06 ± 0.50 aA
EAA	21.81 ± 0.22 a	22.16 ± 0.22 aA	22.51 ± 0.60 aA	21.81 ± 0.22 b	22.70 ± 0.36 aA	22.78 ± 0.01 aA	21.81 ± 0.22 b	22.85 ± 0.02 aA	23.07 ± 0.34 aA
NEAA	39.31 ± 0.47 b	39.75 ± 0.37 bA	41.13 ± 0.21 aB	39.31 ± 0.47 b	41.65 ± 0.57 aA	41.84 ± 0.10 aA	39.31 ± 0.47 b	41.69 ± 0.81 aA	41.99 ± 0.16 aA
E/T(%)	35.68 ± 0.04 a	35.79 ± 0.44 aA	35.37 ± 0.50 aA	35.68 ± 0.04 a	35.28 ± 0.68 aA	35.25 ± 0.05 aA	35.68 ± 0.04 a	35.41 ± 0.43 aA	35.46 ± 0.25 aA
E/N(%)	55.47 ± 0.09 a	55.75 ± 1.07 aA	54.73 ± 1.19 aA	55.47 ± 0.09 a	54.52 ± 1.61 aA	54.44 ± 0.11 aA	55.47 ± 0.09 a	54.83 ± 1.03 aA	54.94 ± 0.59 aA

Note: The data in the table represent the mean ± standard error; the lowercase letters in the same row refer to the significant difference when *p* < 0.05 for different storage periods at the same temperature and the same treatment conditions; the uppercase letters in the same row refer to the *p* values of varying treatment times at the same temperature and the same storage period Significant difference at *p* < 0.05. TAA Total amino acids; EAA; Essential amino acids; NEAA; Non-essential amino acids. E/T(%): Essential/Total); E/N(%): Essential/Non-Essential.

**Table 6 foods-12-01236-t006:** Changes in amino acid composition and content in extruded rice bran during storage.

Amino Acid (mg/g)				Extruded Rice Bran					
	15 °C			25 °C			40 °C		
	0 Day	30 Days	60 Days	0 Day	30 Days	60 Days	0 Day	30 Days	60 Days
Aspartic acid	1.03 ± 0.03 a	0.98 ± 0.05 aA	0.97 ± 0.06 aA	1.03 ± 0.03 a	1.00 ± 0.14 aA	1.02 ± 0.02 aA	1.03 ± 0.03 a	1.01 ± 0.14 aA	1.07 ± 0.07 aA
Threonine	0.74 ± 0.05 a	0.74 ± 0.04 aA	0.65 ± 0.05 aA	0.74 ± 0.05 a	0.64 ± 0.04 a bA	0.61 ± 0.01 bA	0.74 ± 0.05 a	0.41 ± 0.00 bB	0.40 ± 0.06 bB
Serine	0.55 ± 0.01 a	0.49 ± 0.01 bA	0.50 ± 0.03 aA	0.55 ± 0.01 a	0.49 ± 0.04 aA	0.52 ± 0.05 aA	0.55 ± 0.01 a	0.37 ± 0.02 bB	0.27 ± 0.02cB
Glutamic acid	4.61 ± 0.07 a	4.35 ± 0.06 bA	4.45 ± 0.08 a bA	4.61 ± 0.07 a	3.99 ± 0.15 bB	4.09 ± 0.15 bA	4.61 ± 0.07 a	4.13 ± 0.0 bAB	4.14 ± 0.12 bA
Glycine	1.19 ± 0.20 a	1.14 ± 0.05 aA	1.16 ± 0.01 aA	1.19 ± 0.20 a	1.12 ± 0.06 aA	1.09 ± 0.01 aA	1.19 ± 0.20 a	1.03 ± 0.09 aA	1.06 ± 0.10 aA
Alanine	1.28 ± 0.14 a	1.21 ± 0.06 aA	1.25 ± 0.04 aA	1.28 ± 0.14 a	1.18 ± 0.03 aA	1.18 ± 0.00 aA	1.28 ± 0.14 a	1.11 ± 0.12 aA	1.11 ± 0.20 aA
Cystine	0.11 ± 0.01 a	0.12 ± 0.01 aA	0.13 ± 0.02 aA	0.11 ± 0.01 a	0.14 ± 0.01 aA	0.14 ± 0.02 aA	0.11 ± 0.01 a	0.12 ± 0.02 aA	0.12 ± 0.02 aA
Valine	0.63 ± 0.09 a	0.54 ± 0.05 aA	0.56 ± 0.02 aA	0.63 ± 0.09 a	0.57 ± 0.05 aA	0.56 ± 0.07 aA	0.63 ± 0.09 a	0.39 ± 0.03 bB	0.40 ± 0.02 bB
Methionine	0.27 ± 0.02 a	0.25 ± 0.01 aA	0.25 ± 0.02 aA	0.27 ± 0.02 a	0.24 ± 0.00 bA	0.24 ± 0.01 bA	0.27 ± 0.02 a	0.19 ± 0.01 bB	0.20 ± 0.02 bA
Isoleucine	0.36 ± 0.04 a	0.33 ± 0.02 aA	0.35 ± 0.04 aA	0.36 ± 0.04 a	0.35 ± 0.09 aA	0.32 ± 0.07 aA	0.36 ± 0.04 a	0.25 ± 0.03 bA	0.22 ± 0.01 bA
Leucine	1.28 ± 0.15 a	1.04 ± 0.01 aA	1.07 ± 0.04 aA	1.28 ± 0.15 a	1.07 ± 0.08 aA	1.02 ± 0.01 aA	1.28 ± 0.15 a	0.74 ± 0.00 bB	0.79 ±0.07 bB
Tyrosine	0.43 ± 0.02 a	0.40 ± 0.02 aA	0.41 ± 0.03 aA	0.43 ± 0.02 a	0.40 ± 0.03 aA	0.39 ±0.05 aA	0.43 ± 0.02 a	0.36 ± 0.02 bA	0.35 ± 0.01 bA
Phenylalanine	0.27 ± 0.01 a	0.27 ± 0.02 aA	0.27 ± 0.02 aA	0.27 ± 0.01 a	0.28 ± 0.01 aA	0.28 ± 0.02 aA	0.27 ± 0.01 a	0.19 ± 0.01 bB	0.18 ± 0.00 bB
Histidine	1.67 ± 0.01 a	1.43 ± 0.00 bA	1.46 ± 0.05 bA	1.67 ± 0.01 a	1.30 ± 0.00 bB	1.34 ± 0.05 bAB	1.67 ± 0.01 a	1.28 ± 0.03 bB	1.29 ± 0.04 bB
Lysine	1.37 ± 0.09 a	1.21 ± 0.04 aA	1.23 ± 0.02 aA	1.37 ± 0.09 a	1.16 ± 0.02 bA	1.14 ± 0.01 bA	1.37 ± 0.09 a	0.92 ± 0.11 bB	0.89 ± 0.07 bB
Arginine	3.26 ± 0.05 a	2.80 ± 0.06 bA	2.80 ± 0.02 bA	3.26 ± 0.05 a	2.55 ± 0.04 bB	2.53 ± 0.01 bB	3.26 ± 0.05 a	2.49 ± 0.05 bB	2.47 ± 0.07 bB
Proline	0.61 ± 0.03 a	0.63 ± 0.03 aA	0.64 ± 0.00 aA	0.61 ± 0.03 a	0.61 ± 0.25 aA	0.59 ± 0.14 aA	0.61 ± 0.03 a	0.56 ± 0.07 aA	0.56 ± 0.07 aA
TAA	19.67 ± 0.76 aB	17.96 ± 0.05 bA	17.10 ± 0.38 bA	19.67 ± 0.76 aB	18.14 ± 0.05 bB	17.07 ± 0.21 bB	19.67 ± 0.76 aB	15.53 ± 0.20 bC	15.52 ± 0.42 bC
EAA	4.93 ± 0.34 aA	4.41 ± 0.10 aA	4.32 ± 0.21 aA	4.93 ± 0.34 aA	4.37 ± 0.14 aA	4.17 ± 0.15 aA	4.93 ± 0.34 aA	3.08 ± 0.15 bB	3.07± 0.22 bB
NEAA	14.74 ± 0.42 aB	13.55 ± 0.05 bA	12.79 ± 0.17 bA	14.74 ± 0.42 aB	13.77 ± 1.19 bB	12.89 ± 0.06 bB	14.74 ± 0.42 aB	12.45 ± 0.05 bB	12.45 ± 0.20 bB
E/T(%)	25.05 ± 0.77 aB	24.53 ± 0.48 aA	25.23 ± 0.65 aA	25.05 ± 0.77 aB	24.10 ± 0.84 aA	24.45 ± 0.56 aA	25.05 ± 0.77 aB	19.81 ± 0.72 bB	19.79 ± 0.87 bB
E/N(%)	33.43 ± 1.37 aB	32.51 ± 0.84 aA	33.74 ± 1.17 aA	33.43 ± 1.37 aB	31.77 ± 1.47 aA	32.36 ± 0.99 aA	33.43 ± 1.37 aB	24.70 ±1.13 bB	24.67 ± 1.35 bB

Note: The data in the table represent the mean ± standard error; the lowercase letters in the same row refer to the significant difference when *p* < 0.05 for different storage periods at the same temperature and the same treatment conditions; the uppercase letters in the same row refer to the *p* values of varying treatment times at the same temperature and the same storage period Significant difference at *p* < 0.05. TAA Total amino acids; EAA; Essential amino acids; NEAA; Non-essential amino acids. E/T(%): Essential/Total); E/N(%): Essential/Non-Essential.

**Table 7 foods-12-01236-t007:** Linear regression analysis for the Arrhenius plot of rice bran enzymatic activity and fatty acids at varied storage temperatures.

Enzymatic Activity	Temp (°C)	Zero-Order				First-Order				Second-Order			
		k	R^2^	RMSE	X^2^	k	R^2^	RMSE	X^2^	k	R^2^	RMSE	X^2^
PVuntreated	15 °C	0.057	0.841	0.974	0.950	0.020	0.986	0.056	0.00323	−0.004	0.985	0.012	1.48 × 10^−4^
	25 °C	0.059	0.836	1.026	1.053	0.02	0.995	0.030	9.49 × 10^−4^	−0.003	0.969	0.016	2.78 × 10^−4^
	40 °C	0.065	0.843	1.091	1.191	0.023	0.973	0.090	0.008	−0.004	0.893	0.033	0.001
LPS untreated	15 °C	3.514	0.215	261.216	68,233.82	−6.466	0.947	36.160	1307.565	1.88 × 10^−4^	0.975	0.001	1.17 × 10^−6^
	25 °C	3.313	0.201	257.083	66,092.11	−6.495	0.941	38.465	1479.565	2.01 × 10^−4^	0.975	0.001	1.31 × 10^−6^
	40 °C	3.408	0.209	257.874	66,499.10	−6.470	0.964	29.254	855.832	2.14 × 10^−4^	0.952	0.001	2.94 × 10^−6^
FFA untreated	15 °C	2.541	0.967	18.263	333.556	1.988	0.935	12.360	152.793	−0.00174	0.443	0.046	0.002
	25 °C	3.073	0.952	26.776	716.993	2.286	0.890	18.957	359.384	−1.72 × 10^−3^	0.421	0.047	0.002
	40 °C	3.394	0.889	46.520	2164.192	2.047	0.675	3.36 × 10^1^	1128.244	−1.68 × 10^−3^	3.93 × 10^−1^	0.049	0.002
POD	15 °C	0.063	0.758	1.386	1.922	0.009	0.868	0.089	0.008	0.008	0.623	0.256	0.065
	25 °C	0.074	0.807	1.413	1.999	0.019	0.996	0.027	7.36 × 10^−4^	0.007	0.561	0.248	0.061
	40 °C	0.083	0.846	1.388	1.927	0.030	0.975	0.113	0.012	0.006	0.511	0.247	0.061
PV	15 °C	0.060	0.696	1.392	1.939	0.006	0.947	0.036	0.001	−0.001	0.965	0.004	2.12 × 10^−6^
	25 °C	0.062	0.704	1.409	1.988	0.007	0.994	0.013	1.94 × 10^−4^	−0.001	0.987	0.003	9.93 × 10^−6^
	40 °C	0.066	0.731	1.408	1.983	0.011	0.974	0.045	0.002	−0.001	0.954	0.008	7.59 × 10^−5^
LPS	15 °C	0.873	0.830	14.010	196.301	0.334	0.983	1.038	1.077	−3.37 × 10^−4^	0.920	0.002	5.54 × 10^−5^
	25 °C	0.884	0.831	14.145	200.096	0.341	0.980	1.154	1.333	−3.42 × 10^−4^	0.898	0.002	7.41 × 10^−6^
	40 °C	0.903	0.834	14.290	204.227	0.354	0.978	1.254	1.573	−3.46 × 10^−4^	0.888	0.002	8.39 × 10^−6^
FFA	15 °C	0.338	0.694	8.757	76.689	0.001	0.341	0.052	0.002	−7.53 × 10^−4^	0.341	2.47 × 10^−4^	6.12 × 10^−8^
	25 °C	0.398	0.783	8.150	66.436	0.089	0.639	1.582	2.503	−3.12 × 10^−4^	0.644	0.005	3.00 × 10^−5^
	40 °C	0.397	0.783	8.151	66.4470	0.088	0.640	1.578	2.490	−3.12 × 10^−4^	0.645	0.005	2.98 × 10^−5^

PV denotes Peroxide value; LPS: Lipase activity; POD: Peroxidase activity; FFA: Free fatty acids.

## Data Availability

The data presented in this study are available on request from the corresponding author.

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
