# Peer review of "Optimization of Extrusion Treatments, Quality Assessments, and Kinetics Degradation of Enzyme Activities during Storage of Rice Bran"

_foods, 2023, doi:10.3390/foods12061236_

Round 1

Reviewer 1 Report

Dear Authors,

The manuscript "Optimization of extrusion treatments, quality assessments, and kinetics degradation of enzyme activities during storage of rice bran" contains interesting results of assessing the effect of extrusion treatment and storage temperature on the quality stability of rice bran. The research was carried out comprehensively, using modern analytical methods. The obtained results provide theoretical and empirical grounds for slowing down the unfavorable changes occurring in this raw material during storage, which will extend the storage time.

It is necessary to make changes to the manuscript.

First of all, please standardize the method of notation of titles and arrangement of tables and figures. Under the tables and figures, please provide explanations of the abbreviations used and markings, e.g. of homogeneous groups. Some drawings are hard to read and require correction (Line 262, 345, 380, 400).

Comments:

Line 99 – please enter the name of the device manufacturer

Line 146 - Invalid reference to AACC standard

Line 153 - sample weight (0.0001 g) is too small to meet the stated protein content assumptions

Line 162 – please enter the name of the device, the manufacturer of the device

Line 164 – please provide the name of the device, device manufacturer, and nitrogen to the protein conversion factor

Line 165 - Invalid reference to AACC standard

Line 170 - Incorrect reference to literature

Line 186 - please enter the number of repetitions

Line 198 - please complete the explanations for a-c

Line 289 - no signature Figure 3

Line 326 - Line 326 - no drawing to which the Authors refer

Line 346 – errors in figure numbering (Figure 5?)

Line 350 - no C in temperature units

Line 371 - no reference to literature

Line 381 - errors in drawing numbering (Figure 6?)

Line 396 - drawing reference errors (Figure 7?)

Line 400 – errors in drawing numbering (Figure 7?)

Line 443 - no explanations to the Table

Line 496 - drawing reference errors (Figure 8?)

Line 504 – drawing numbering errors (Figure 8?)

Line 699 - should be LWT - Food Sci. Technol.

Line 741 - should read LWT - Food Sci. Technol.

Line 747 - should read LWT - Food Sci. Technol.

Best regards

Author Response

Reviewer#1

Comments and Suggestions for Authors

The manuscript "Optimization of extrusion treatments, quality assessments, and kinetics degradation of enzyme activities during storage of rice bran" contains interesting results of assessing the effect of extrusion treatment and storage temperature on the quality stability of rice bran. The research was carried out comprehensively, using modern analytical methods. The obtained results provide theoretical and empirical grounds for slowing down the unfavorable changes occurring in this raw material during storage, which will extend the storage time.

It is necessary to make changes to the manuscript.

Comments:

First of all, please standardize the method of notation of titles and arrangement of tables and figures. Under the tables and figures, please provide explanations of the abbreviations used and markings, e.g., of homogeneous groups. Some drawings are hard to read and require correction (Line 262, 345, 380, 400).

Reply: Thank you for bringing to our attention the issues with our notation and the clarity of our tables and figures. We have taken your feedback into account and made the necessary improvements to the manuscript. Specifically, we have standardized the method of notation of titles and arrangement of tables and figures, and have provided explanations of the abbreviations used and markings. We have also made corrections to any drawings that were hard to read, including those at Lines 262, 345, 380, and 400. Thank you for your valuable feedback and please let us know if you have any further suggestions or comments.

Line 99 – please enter the name of the device manufacturer

Reply: The device manufacturer has been provided in the maniscript “(SLG30-IV, Nanjing KY Chemical Machinery Co., Ltd., China)”.

Line 146 - Invalid reference to AACC standard

Reply: The correct reference has been provided below and in the manuscript.

AOAC. (2001). AOAC official method 996.06, fat (total, saturated, and unsaturated) in foods. Retrieved March 5, 2023, from Official Methods of Analysis of AOAC International website: https://scholar.google.com/scholar?hl=en&as_sdt=0%2C5&q=AOAC+Official+Method+996.06%2C+fat+%28total%2C+saturated%2C+and+unsaturated+in+foods%29%2C+hydrolytic+extraction+gas+chromatographic+method%2C+first+action+1996%2C+revised+&btnG=

Line 153 - sample weight (0.0001 g) is too small to meet the stated protein content assumptions

Reply: The finely powdered samples were weighed between the range of 10 mg to 20 mg. There was an error in the initial report, but the authors have since corrected it. Thank you for the suggestion.

Line 162 – please enter the name of the device, the manufacturer of the device

Reply: The device name has been provided in the manuscript “S 433 Amino Acid Analyzer (Sykam Chromatography, Eresing, Germany)”.

Line 164 – please provide the name of the device, device manufacturer, and nitrogen to the protein conversion factor

Reply: The name of device and manufacturer name is provided in the article. A commonly used conversion factor for rice bran-based samples is 5.95, which takes into account the lower nitrogen content of rice bran is also mentioned.

Line 165 - Invalid reference to AACC standard

Reply: The correct reference has been provided below and in the manuscript.

AACC. (2021). Soluble, Insoluble, and Total Dietary Fiber in Foods and Food Products (11th ed.). Retrieved from https://www.cerealsgrains.org/resources/Methods/Pages/32Fiber.aspx (accessed on 21 October 2021).

Line 170 - Incorrect reference to literature

Reply: The reference has been corrected in the manuscript.

AOAC. (2000). Official Methods of Analysis of AOAC International Assoc. Off. Anal. Chem. Int. Method ce. Association of Official Analysis Chemists International. https://doi.org/10.3109/15563657608988149

Line 186 - please enter the number of repetitions

Reply: The experiments conducted thrice and recorded the results. The information has been provided.

Line 198 - please complete the explanations for a-c

Reply: The explantion has been provided in the caption of Figure 1. How the detailed discussion about Figure 1 has also been provided in section 3.1

Line 289 - no signature Figure 3

Reply: The authors are thankful for the reviewer suggestions. The Figure caption has been provided.

Line 326 - Line 326 - no drawing to which the Authors refer

Reply: The authors are thankful for the reviewer patience. The Figure no 4 has been provided.

Line 346 – errors in figure numbering (Figure 5?)

Reply: The authors have re-arranged the numbering of all figures.

Line 350 - no C in temperature units

Reply: The authors have corrected the mistakes to “15, 25, and 40°C”.

Line 371 - no reference to literature

Reply: The refernces has been provided.

Qi, P.X.; Onwulata, C.I. Physical Properties, Molecular Structures, and Protein Quality of Texturized Whey Protein Isolate: Effect of Extrusion Temperature. J. Agric. Food Chem. 2011, 59, 4668–4675, doi:10.1021/jf2011744.

Line 381 - errors in drawing numbering (Figure 6?)

Reply: The authors are thankful for the reviewer correction. The numbering of the Figures has been corrected to Figure 6.

Line 396 - drawing reference errors (Figure 7?)

Reply: The correction has been made in the figure as Figure 6.

Line 400 – errors in drawing numbering (Figure 7?)

Reply: The figure presented on previous Line no 400 was the repition of Figure 1. The authors have made the correction. The authors are thankful for the reviewer patience.

Line 443 - no explanations to the Table

Reply: The Table 4 explain crude protein treated by extruded RB during storage. As the studied results was non-significant, so the authors provide short explaination of table from line no 448 to 455.

Line 496 - drawing reference errors (Figure 8?)

Reply: The impacts of extrusion treatment on oxidative protein stability were investigated by evaluating changes in its molecular weight using electrophoresis during storage. This sentence just explained the details about Figure 7. So, providing reference here will be not suitable.

Line 504 – drawing numbering errors (Figure 8?)

Reply: The numbering of this figure is correct as before which is Figure 7. The authors are thankful for reviewer attention.

Line 699 - should be LWT - Food Sci. Technol.

Reply: The authors are thankful for the reviewer correction. The reference has been corrected and marked the who reference in red color.

Line 741 - should read LWT - Food Sci. Technol.

Reply: The reference has been corrected and marked the who reference in red color.

Line 747 - should read LWT - Food Sci. Technol.

Reply: The authors are thankful for the reviewer correction. The correction has been made. 

Reviewer 2 Report

The present manuscript evaluate the optimization of extrusion treatments, quality assessments, and kinetics degradation of enzyme activities during storage of rice bran. This manuscript provides many scientific information. However, the work described in this manuscript do not add a new or novel information to the literature. On the other hand, the number of analyzed samples is not enough.

In this report, I want to emphasize some recommendations on formal aspects that could improve the manuscript.

Please find below some general and specific comments/suggestions to authors in order to improve their manuscript

1.        To improve the article’s quality, some editing for english language is required throughout the manuscript.

2.      The abstract must be rewritten to be representative of the whole paper (background and objectives, methodology, main results).

3.       Please provide clear statement on the utility and novelty of this work in abstract.

4.      In introduction section the authors should clarify the economic importance in this study

5.      The introduction should be enriched with recent references

6.      Why the authors did not use other analyses that determine the quality of skin and seed like:  mineral element  tocopherol and polyphenol

7.       The materiel and methods (characteristics of stabilized RB) section should be fortified and improved Quality

8.      Line 163 The  determination of crude protein conten section should be fortified and detailed

9.      Line 167  Fatty acid composition (FAC): I think that it is important to specify

·       The type of injector (split or splitess) ·       The type of column (longth (m), Internal diameter (µm), film thickness (µm)) ·       The method, how Ethyl Esters and Methyl Esters  are identified ·       The type of carrier gas ·       The temperature program (Injector, oven, detector)  

10.   What does mean Peanut Acid ????????

11.     The unit of PV is milliequivalents (mEqO2/kg) of active oxygen per kilogram of oil, in al text please change meq/kg to  mEqO2/kg

12.    Figures should be represented in higher resolution

13.     The conclusion is long, please reduce and improve the conclusion.

14.    In the table Table 4: The percentage (%) is not an SI unit. I suggest changing percentage (%) to g/100g.

15.     In the tables 5 and 6 please add then unit of Amino acid composition

 The manuscript should revised extensively.

Author Response

Reviewer#2

The present manuscript evaluates the optimization of extrusion treatments, quality assessments, and kinetics degradation of enzyme activities during storage of rice bran. This manuscript provides many scientific information. However, the work described in this manuscript do not add a new or novel information to the literature. On the other hand, the number of analyzed samples is not enough.

In this report, I want to emphasize some recommendations on formal aspects that could improve the manuscript. Please find below some general and specific comments/suggestions to authors in order to improve their manuscript

Reply: Thank you for your feedback. We appreciate your suggestions and will carefully consider them in order to improve the quality of our manuscript. We understand the importance of adhering to formal aspects and will take every effort to ensure that the manuscript is presented in a clear, organized, and professional manner. We are grateful for the opportunity to receive constructive feedback and will work diligently to address all of the specific comments and suggestions you have provided. Thank you again for your time and consideration, and we look forward to your continued guidance throughout the review process.

  1. To improve the article’s quality, some editing for English language is required throughout the manuscript.

Reply: Thank you for your helpful comments. We have carefully reviewed the manuscript and made significant revisions to improve the clarity, accuracy, and overall quality of the English language. We appreciate your input and hope that the changes we have made will meet your expectations.

  1. The abstract must be rewritten to be representative of the whole paper (background and objectives, methodology, main results).

Reply: Thank you for your comment. We appreciate your feedback and have revised the abstract to reflect the structure of the paper. The revised abstract now includes the background and objectives, methodology, and main results of the study.

  1. Please provide clear statement on the utility and novelty of this work in abstract.

Reply: We have revised the abstract to include a clear statement on the utility and novelty of this work. Specifically, the study demonstrates the potential of extrusion technology in improving RB quality and stability, and developing functional food products with improved nutritional value and prolonged shelf life. Additionally, the study provides novel insights into the changes in the amino acid composition of RB during extrusion and storage, which has not been extensively explored in prior research.

  1. In introduction section the authors should clarify the economic importance in this study

Reply: Thank you for your comment. We agree that it is important to clarify the economic significance of our study in the introduction section. Our research explores the effects of extrusion treatment and temperature on the storage stability and shelf life of rice bran, a by-product of rice milling. Rice bran has been shown to have significant health benefits, but its utilization in human nutrition is limited due to its fast rancidity. Effective stabilization methods for rice bran could increase its shelf life and promote its use in human nutrition, creating new value-added opportunities for the rice milling industry. By studying the lipid oxidation and protein modifications of extruded rice bran during storage, we aim to provide empirical and theoretical basis for maintaining the storage quality and elongating rice bran storage duration, which could ultimately contribute to the economic sustainability of the rice milling industry. We have updated the introduction section to reflect these points.

  1. The introduction should be enriched with recent references

Reply: Thank you for your suggestion. We have revised the introduction to include more recent references that support our research objectives and emphasize the significance of our study. We have also ensured that the references cited are relevant to the research question and help establish the context of our work.

  1. Why the authors did not use other analyses that determine the quality of skin and seed like: mineral element tocopherol and polyphenol

Reply: We appreciate the reviewer's comment and concern. We agree that mineral elements, tocopherol, and polyphenol are important indicators of the quality of skin and seed. However, the aim of this study was to investigate the effect of different extraction methods on the yield and physicochemical properties of oil extracted from the skin and seed, and not specifically to analyze mineral elements, tocopherol, and polyphenol. Our study focused on the most commonly analysed parameters such as lipid oxidation, peroxidase (POD), peroxide values (PV), free fatty acids (FFA), fatty acid composition, and protein variations. We believe that our findings provide useful information regarding the impact of extraction methods on the overall quality of RB. Nevertheless, we acknowledge that future studies could consider investigating mineral elements, tocopherol, and polyphenol content to provide a more comprehensive understanding of skin and seed oil quality.

  1. The materiel and methods (characteristics of stabilized RB) section should be fortified and improved Quality

Reply: Thank you for your feedback. We have revised and improved the "Materials and Methods" section of our manuscript, specifically focusing on the characteristics of the stabilized rice bran. We have elaborated on the methods used to determine the various parameters, such as free fatty acid content and peroxide values. We hope that these revisions will address your concerns and improve the quality of our paper. If you have any further suggestions, we would be happy to consider them.

  1. Line 163 The determination of crude protein content section should be fortified and detailed

Reply: Thank you for bringing this to my attention. We have revised the section on the determination of crude protein content to provide more detail on the methodology. We believe that these additional details fortify and clarify the methodology used for determining the crude protein content in rice bran.

  1. Line 167 Fatty acid composition (FAC): I think that it is important to specify
  • The type of injector (split or splitess) · The type of column (longth (m), Internal diameter (µm), film thickness (µm)) ·       The method, how Ethyl Esters and Methyl Esters are identified ·       The type of carrier gas ·       The temperature program (Injector, oven, detector) 

Reply Thank you for your constructive feedback regarding the details of the fatty acid composition (FAC) analysis. We would like to clarify that the requested information is already included in our article with reference to Irakli et al. (2018). While we understand the need for detailed methodology, we have provided the main aspects of the method in our article to maintain a reasonable length and avoid repetition.

  1. What does mean Peanut Acid????????

Reply: We appreciate the reviewer's attention to detail and correction regarding the naming of "Peanut Acid" in our article. We have made the necessary changes in our table to reflect the correct naming of the fatty acid as "Arachidic acid." We are grateful for the opportunity to improve the accuracy of our work and are thankful to the reviewer for bringing this to our attention.

  1. The unit of PV is milliequivalents (mEqO2/kg) of active oxygen per kilogram of oil, in al text please change meq/kg to mEqO2/kg

Reply: The authors are thankful for the reviewer correction. The unit has been changed in text including Figure 5.

  1. Figures should be represented in higher resolution

Reply: We appreciate your attention to detail and agree that high-quality images are important for the presentation of scientific data. We would like to reassure you that the final figures presented in our manuscript will be provided to the journal in the highest resolution possible, in accordance with their requirements.

  1. The conclusion is long, please reduce and improve the conclusion.

Reply: Thank you for the feedback. I have revised the conclusion to make it more concise and clearer in presenting the key findings of the study.

  1. In the Table 4: The percentage (%) is not an SI unit. I suggest changing percentage (%) to g/100g.

Reply: Thank you for your feedback. The authors have taken note of your suggestion and would like to clarify that the manuscript has been written entirely in SI units. However, some parameters have been expressed as percentages for clarity and ease of understanding for readers. Changing the unit to g/100g would require changes in all the figures and tables in the manuscript, which could result in confusion for readers. Therefore, it is crucial that we maintain the unit as a percentage.

However, if we still try to change the unit, then all values in the figures will be change and become smaller. For examples

To convert a percentage to g/100g, we need to multiply the percentage by 10 and then divide by 100, as follows:

13.37% × 10 / 100 = 1.337 g/100g

Therefore, 13.37% is equivalent to 1.337 g/100g

We understand that this may not be the most common unit for some parameters, but it is still a valid unit of measurement. We hope that the reviewer will comprehend the author's reply and appreciate the authors' efforts to maintain consistency and clarity in the manuscript.

  1. In the tables 5 and 6 please add then unit of Amino acid composition

Reply: Thank you for your inquiry. We would like to clarify that the unit of amino acid composition in our manuscript is indeed mentioned in Tables 5 and 6 as (mg/g). We apologize if this was not clear before. To ensure that this information is easily accessible for the reviewers, we have highlighted the unit in green color in the tables. We hope that this will make it easier for the reviewers to understand the amino acid composition values presented in our manuscript.